# A Conceptual Model Based on the Activity System and Transportation System for Sustainable Urban Freight Transport

**Demostenis Ramos Cassiano** [1,*], **Bruno Vieira Bertoncini** [1] **and Leise Kelli de Oliveira** [2]

1 Postgraduate Program in Transport Engineering, Transport, Traffic and Environment Research Group, Federal University of Ceará, Fortaleza 60455-760, Brazil; bruviber@det.ufc.br
2 Department of Transportation and Geotechnical Engineering, Universidade Federal de Minas Gerais, Belo Horizonte 31270-010, Brazil; leise@etg.ufmg.br
* Correspondence: cassiano@det.ufc.br

**Abstract:** Urban freight transport (UFT) is simultaneously responsible for maintaining the urban lifestyle and the negative externalities impacting urban areas, necessitating strategies that promote sustainable urban freight transport (SUFT). In addition, the stakeholders and geographic factors involved in UFT impose specific concerns in the planning and operation stages of SUFT. Therefore, this paper proposes a model addressing sustainable last-mile delivery considering the relationship between the activity system, transportation system, and stakeholders involved in UFT. Based on the literature review, we identified UFT planning procedures to achieve SUFT. In a cyclical process, these procedures were considered on the proposed model, integrating freight transport planning with urban planning to develop SUFT and, consequently, sustainable cities.

**Keywords:** sustainable urban freight transport; last-mile delivery; urban activity system; transport system; sustainable city; stakeholders

## 1. Introduction

The importance of urban freight transport (UFT) to maintain the daily lives of city residents contrasts with the negative externalities of this system [1–6]. Considering this importance, UFT has been the object of studies by academics, technicians, and managers. The complexity of UFT is driven by factors such as technological, geographical, and cultural features, as well as the diversity of products, aggregation units, vehicle types, and stakeholders' objectives and needs [3,4,7–10].

The growing volume of goods transported increases transport network use by freight vehicles [11,12], negatively impacting the urban environment. The reduction of these externalities and the improvement of mobility and quality of life make the inclusion of UFT in city strategic management necessary [13,14]. However, this inclusion has challenges, mainly related to stakeholders' knowledge about the UFT phenomenon. Stakeholders know their own objectives; however, the interaction of their objectives with the objectives of other stakeholders is unknown. We assumed that understanding the relationship between UFT, the activity system, and the transportation system could improve the knowledge about the UFT phenomenon, creating a cooperative and collaborative environment involving all stakeholders to achieve SUFT. Therefore, the understanding of the UFT phenomenon still presents a research gap.

UFT is a dynamic phenomenon with specificities linked to characteristics, such as geographical and cultural factors. UFT dynamism is mainly related to economic and operational factors integrated into technological development (e.g., an intelligent transportation system). Geographic and cultural factors contribute to scenarios associated with UFT. In this way, business-to-business (B2B) and business-to-consumer (B2C) relationships are beyond UFT solutions [6,7,15–21].

Some of the externalities arising from the UFT operation are congestion, pollution—visual, noise, and air—decreased road safety, and infrastructure damage. Recently, the literature has focused on UFT solutions to mitigate externalities [1–5,22–25]. However, the UFT solution depends on understanding the phenomenon and an approach that covers its specificities.

UFT problems are intrinsically contained in the relationship between its operation and the city, represented by the relation between the urban activity system (AS) and the transportation system (TS). Understanding the UFT phenomenon drives the following research question: How are the urban AS and TS related to UFT?

In general, UFT literature focuses on vehicular routing to reduce delivery costs. However, in recent years, SUFT, the influence of the urban environment, and concern about the growth of e-commerce have become important in the literature. In this context, this paper presents a conceptual model addressing SUFT, considering the relationship among the AS, TS, and stakeholders involved in UFT to develop a holistic view of sustainable cities. The conceptual model was based on the literature identifying planning procedures for developing SUFT. The innovation is related to the integration of UFT planning in urban planning to develop sustainable cities. Understanding the UFT phenomenon from the relationship between AS and TS allows us to minimize the problems arising from the growing demands of stakeholders, including the operational innovations in the transport sector, and makes the sustainable urban planning process more efficient.

The conceptual approach contributes to improving the stakeholders' knowledge about UFT, its externalities, challenges, and opportunities for a sustainable future, promoting a collaborative and participative environment for all stakeholders.

This paper has six sections, including the introduction. Section 2 describes the research method based on the literature review. The literature review to identify the evolution from city logistics measures to SUFT is presented in Section 3, showing the planning procedure and its interaction. Section 4 introduces and discusses the conceptual model. Section 5 discusses the advantages and limitations of the model presented in Section 4, and Section 6 shows the practical implications of the conceptual model and presents conclusions.

## 2. Research Method

The research method was based on a literature review to answer how the urban AS and TS relate to UFT. We used Google Scholar as a database and the following set of keywords (urban planning, last-mile deliveries, urban logistics, city logistics, UFT, sustainable logistics, and SUFT) to identify the current state of knowledge about UFT planning and how the literature relates UFT, AS, and TS.

We identified the main stages for proposing the UFT planning procedure from reading the articles and systematizing the knowledge. In addition, we associated a cyclical process among the stages of the planning procedure, creating the relationship between planning procedures. Finally, we distinguished how UFT is related to the AS and TS to propose a conceptual model that integrates both, including stakeholders' objectives.

## 3. From City Logistics to SUFT—A Literature Review

City logistics seek to mitigate the UFT externalities by considering the concept of sustainable development [26]. In this sense, environmental, social, and economic development become objectives for SUFT. UFT complexity is the result of the goals and requirements of its stakeholders. UFT's externalities result from its unbalanced consumption of resources caused by a lack of UFT planning and SUFT operation [2–5].

City logistics are measures to improve UFT efficiency, respond to customer demands, and reduce UFT externalities. [27,28]. To achieve SUFT, those involved in UFT planning need to (1) understand UFT by making a diagnosis of the local situation; (2) identify the stakeholders and their goals, (3) identify operational solutions addressed to UFT-related environmental impacts, (4) identify technological solutions and (5) last-mile solutions,(6) model these measures to evaluate their impacts, (7) propose a business model to enhance

the operational, economic, and social feasibility, and, finally, (8) implement an urban freight public policy based on the results of this procedure. These steps are presented in Table 1.

**Table 1.** Planning procedures and related literature to achieve SUFT.

| Planning Procedures | Related Literature |
|---|---|
| (1) Understanding UFT | [29–35] |
| (2) Stakeholders and their relations | [10,33,36–40] |
| (3) Operational solutions addressed to environmental impacts | [2,9,41–47] |
| (4) Technological solutions | [16,48–57] |
| (5) Last-mile solutions | [29,58–69] |
| (6) UFT modelling | [2,7,8,15,27,70–72] |
| (7) Business model | [19,20,73,74] |
| (8) Development of urban freight public policies | [3,4,75–79] |

Understanding UFT is the first city logistics procedure. The recognition of the UFT process allows for identifying gaps and related measures; for this, data is essential. According to Campagna et al. [35], the lack of data for a diagnosis of freight flows is one of the main obstacles to UFT planning, operational efficiency, and achieving SUFT. For Pattier and Routhier [80], the goods movements obtained from the establishment of driver surveys could describe the city's vehicle flow. Allen et al. [81] presented the most common survey techniques, including stakeholders' surveys. Oliveira [82] proposed a framework with UFT data. With technological advancements, floating car data became typical to identify delivery routes' [8]. In addition, UFT diagnosis could be obtained from GPS data [83], mobile applications [84], and big data [85], among others. Allen et al. [86] pointed out the main gaps in the data collection, such as vehicle activity, supply chain, freight and logistics infrastructure, loading and unloading operations, and the data about non-road modes.

The identification of stakeholders and their relationships is considered in procedure 2. In this way, recognizing the stakeholders' heterogeneity is a meaningful way to include them in all stages of the planning process by sharing a continuous flow of information and preserving the specifics of their perceptions. Alho et al. [10] analyzed the stakeholders' heterogeneity and proposed a framework to represent the UFT phenomenon and its interactions in all stages of the planning process by sharing a continuous flow of information. The stakeholders' perceptions of freight policies were analyzed by Amaya et al. [37] in two Colombian cities. Considering carriers, receivers, and citizens, the authors showed a unique perspective from some solutions in a geographical area and a diverse perspective in other geographical locations, showing the importance of geographical attributes for city logistics measures. Oliveira and Oliveira [39] analyzed the perception of carriers, retailers, residents, and public managers for city logistics solutions in Belo Horizonte, Brazil. In general, carriers and local authorities had similar perceptions, while the retailers had opposite points of view. The residents had a positive perception, while the solution implied mobility improvements.

Procedures 3 through 5 address city logistics measures. Procedure 3 focuses on operational solutions, such as off-peak delivery, off-night delivery, and unloading bays, to reduce environmental impacts [2]. The operational solutions improve freight mobility and reduce delivery time and emissions [87–89]. Unloading bays are the most common solution for SUFT [90], contributing to improved mobility [91]. Establishment data could support unloading bay analysis [91], while operational tests could show off-peak delivery or off-night delivery benefits.

Procedure 4 focuses on technological solutions based on the intelligent transportation system (ITS). The ITS provides real-time information to support vehicle routing and scheduling procedures [92] and vehicle movements [93]. The ITS classification in freight transport was shown in Gattuso and Pellicano [94], while Crainic et al. [16] evaluated freight ITS. Małeckia et al. [51] showed that ITS could contribute to mitigating UFT-related environmental impacts. ITS, information and communication technology, the internet of

things, big data, and artificial intelligence could contribute to SUFT [49]. Technological solutions, such as parcel boxes, delivery boxes, collection and delivery points, or lockers related to last-mile deliveries, have been developed in recent years. Electric vehicles and autonomous driving vehicles also add to the technological solutions, an essential step to SUFT [53].

Procedure 5 considers last-mile solutions. The growth of e-commerce has directly affected the evolution of last-mile deliveries, mainly home deliveries. Dispersed urban deliveries increase the environmental impact, challenging the logistical process of companies for deliveries within a short time window. Pickup points, parcel lockers, click-and-collect systems, or the collection and delivery points [95] are solutions that are becoming necessary to reduce last-mile delivery times in urban densified areas. Janjevic and Winkenbach [60] proposed a framework that characterized the last-mile solution, showing the variables influencing the last-mile strategy's choice. Oliveira et al. [95] presented a theoretical basis concerning the collection and delivery points and evaluated the importance of accessibility to these points to reduce home delivery times. The last-mile solution addressing the HORECA (hotellerie-restaurant-café) sector is essential for optimizing the logistics operation [62]. Literature concerning last-mile deliveries is vast and considers on-demand deliveries [38,58,59,96], electric vehicles [63–67,97,98], and cargo bicycles [64–66,68,99].

On-demand deliveries are increasing with digital platforms, which require instant deliveries [100]. Bjørgen et al. [58] evaluated the impacts of e-grocery shopping and how city municipalities integrate urbanization and digitalization in sustainable urban planning to improve freight mobility. Founded on the sharing economy, crowd logistics is another last-mile solution related to on-demand deliveries [96]. Rai et al. [96] identified crowd logistics initiatives and the respective stakeholders involved. Many on-demand deliveries could negatively impact fuel consumption and emissions due to the vehicle type used [31,101]. While electric vehicles reduce fuel consumption and emissions, cargo bicycles reduce travel time and emissions [99,102]. The cargo bicycle is a simple and cheap last-mile solution, while electric vehicles have challenges such as battery life, a sustainable energy source, and vehicle cost [63].

Since the UFT problems have been characterized, the modeling is essential to evaluate the local context's impacts. Procedure 6 considers UFT modeling. Most of the literature brings some approach addressed to measure UFT impacts and evaluate freight policies. As an essential part of the UFT planning process, the modeling needs to support ex ante assessment of freight policies considering the planning horizon and all stakeholders. Most UFT modeling approaches focus on freight demand [7], delivery routes [8], and ex ante evaluation [72,103]. Considering the stakeholders, Kaszubowski [104] proposed a method for local authorities to evaluate UFT. Comi et al. [7] proposed a methodology-based four-step model considering land use transport interaction (LUTI) and AS. Since UFT is dynamic, the four-step method cannot identify this dynamism, despite LUTI and AS inclusion. For this reason, the holistic view is essential for UFT modeling, mainly in the evaluation of city logistics measures.

Procedure 7 deals with a business model, i.e., how the company can create, deliver, and capture financial, social, cultural, and environmental value [18]. Quak et al. [20] discussed the business factors for city logistics measures by analyzing the business model for the Bentobox. For Björklund et al. [19], the business model related to UFT is complex due to business factors and social factors. Still, the complexity of UFT does not allow for the design of a standard business model, which depends on the city's characteristics.

To identify the feasibility of the solution, procedure 8 focuses on urban freight public policies, which consider all planning steps to propose freight policies evaluated by indicators to verify the achievement of the goals proposed. A public policy effectively addresses common problems identified in a participatory process [105]. Policymakers seek to balance UFT efficiency and environmental sustainability, but the conflicting objectives of stakeholders bring more complexity to this process [106]. Paddeu and Aditjandra [107] proposed a participatory approach to identify problems and related solutions in an engaging and easy

process to develop public policies. As the measures depend on and change with the local characteristics, the freight policies need to be adapted to represent reality.

The understanding of planning procedures leads to a cyclical process, and its execution contributes to the development of UFT public policies, as presented in Figure 1.

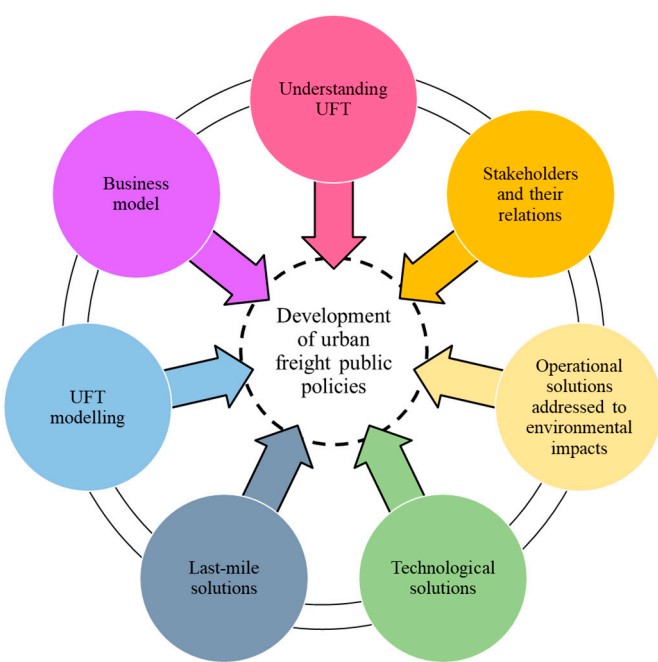

**Figure 1.** The relationship between planning procedures.

Public policies must consider solutions with positive urban mobility impacts (from the perspective of public authorities) while contributing to new businesses (from the private sector's perspective). Thus, the planning process must be results-oriented and monitored by indicators in a continuous monitoring and evaluation process to enhance sustainable freight mobility. However, the inclusion of UFT in public policies is a challenge for public authorities. A conceptual model to support knowledge about the UFT and its relation to the AS and TS could mitigate the challenges related to interactions among stakeholders and UFT's objectives. This conceptual model is proposed in the next section.

## 4. A Conceptual Model for Sustainable Urban Freight Transport

Figure 2 presents the conceptual model for SUFT based on the AS and TS relationship in an urban area and its consequent impacts on UFT. UFT occurs in urban areas, which include AS and TS. The operation of UFT is influenced by environmental, economic, and social factors measured and evaluated by indicators. In addition, the impact measures (e.g., fuel consumption, tax, and travel time, among others) affect stakeholders' opinions, which shift the operation to enhance the indicators' improvements and, consequently, achieves SUFT and contributes to sustainable cities.

The relationship among AS, TS, impact measures, and stakeholders could promote SUFT. The impact measures are a way to quantify the effects of UFT on the urban environment and are directly related to the stakeholder's objectives. For example, an e-consumer buys a product. This purchase takes place in the AS. The purchase generates a freight trip, which occurs in the TS, represented by the cargo flow. The cargo flow generates information flow, which will allow estimating the impact measures. Considering the impact measures, the stakeholders define strategies to achieve SUFT. These strategies change the AS and TS, influencing the UFT operation and, consequently, the impact measures. The impact measures are evaluated by indicators, as explained hereinafter. The impact measures allow for the evaluation of the effectiveness of the SUFT policies. The inclusion

of new technologies and innovative solutions, such as crowd logistics [21,59,96], electric vehicles [55,57,63], and autonomous guided vehicles (AGV) [49,52,54], change the last-mile operation and have the potential to accelerate the benefits of SUFT.

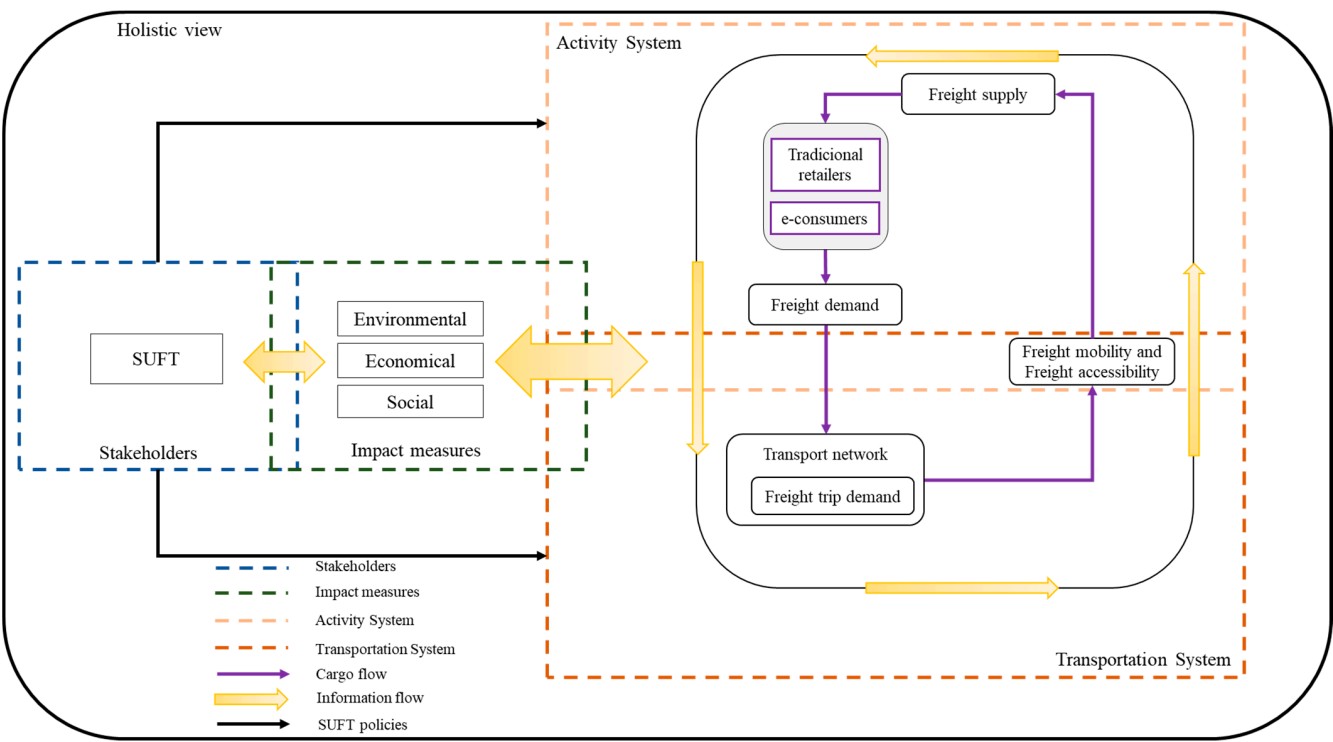

**Figure 2.** A conceptual model for sustainable urban freight transport.

### 4.1. Urban Area

The last-mile deliveries occur in urban areas. Retailers or individual consumers make a request that generates a demand on the AS and TS. According to Lopes et al. [108], the activities that occur in the city are essential for location decisions and mobility patterns.

Among the concepts for the AS in urban areas, Coppola and Nuzzolo [109] define it as a social and economic pipeline. Additionally, Arentze and Timmermans [110] describe the AS as personal needs or desires distributed in time and space. Thus, it is natural to understand that the AS includes those directly related to UFT, such as the shops [31,109,111]. However, e-commerce brings another virtual side for the AS, creating a direct link for the e-consumers. Therefore, the system activities are composed of the traditional retailers and e-consumers, requiring freight demand and freight trip demand in an urban freight system. The requirements of conventional retailers and e-consumers are different concerning time, area, and consumer relations, bringing complexity to the last-mile delivery.

The TS comprises products, transportation modes, and the transportation infrastructure, which allows freight mobility [112] to supply the freight demand and freight trip demand [113]. In particular, the transportation infrastructure system could improve UFT [3,114–116].

Freight demand is related to the AS. Freight trip demand is associated with the TS. Freight mobility and freight accessibility occur in the TS. Freight mobility is the way to move products to their final destination (retailers or e-consumers). Freight accessibility is a critical issue to achieve the final destination, and it depends on the availability of unloading bays and street connections.

In a simplified way, retailers or e-consumers request freight demand in an AS. Freight demand requires freight trip demand, which will occur in the TS. Freight mobility and freight accessibility could provide efficient last-mile delivery. However, e-commerce brings

complexity to last-mile delivery [29,58,95] due to sparse deliveries in a geographic region. In addition, the delivery cost, delivery time, product availability, and travel distance, among others, could influence customer shopping behavior (traditional retailer or e-commerce). This cargo flow is directly affected by the desires of UFT stakeholders, as presented in the following subsection.

### 4.2. Stakeholders

The main stakeholders involved in UFT are those who request, dispatch, transport, and receive [23,31,117–119]. Table 2 details these stakeholders, their function, and objectives.

**Table 2.** Stakeholders: functions and objectives.

| Stakeholders | | Function | Objective |
|---|---|---|---|
| Who requests | Retailers | Offer products | Profit from supplying products |
| | E-consumers | Consume products | Meet personal needs for products |
| Who dispatches | Shippers | Send the requested goods to receivers | Send cargo to meet the receivers wishes |
| Who transports | Carriers | Transport products | Profit by transporting goods |
| Who receives | Retailers | Contribute to economic development | Receive products |
| | E-consumers | | |
| Who promotes | Local authorities | Promote a sustainable city | Promote a sustainable city, including UFT |
| Who resides | Residents | Reside in a sustainable city and consume the available products | Live in a city with livability and access to desired products |

The UFT challenges are related to identifying the stakeholders and the relationships between them [120]. The harmony among the stakeholders becomes essential for the development of sustainability in the cities. The freight quality partnership (FQP) is one way for local authorities to include stakeholders in the discussion to develop sustainable freight solutions, enhancing the identification of success factors to satisfy stakeholders' needs [121,122]. Kijewska and Jedliński [122] proposed the pyramid of stakeholders, which divides stakeholders into three groups: residents (demand for goods), businesses (market response for the order), and local authorities (responsible for implementing city logistics measures). An FQP involving these groups could enhance the maturity and durability of SUFT projects [122]. Lindholm [121] presented the way to establish FQP, considering the Gothenburg case. Bjørgen et al. [123] showed the criteria to ensure the stakeholders' participation in a collaborative process.

### 4.3. Impact Measures

UFT planning needs to include the UFT impacts [124,125], which are crucial for evaluating the achievements of SUFT [62,119,120,126,127].

The most negative environmental impacts from UFT come from emissions and ineffective energy consumption [128–131]. The economic impacts are related to distribution cost [101], vehicle type [30], and infrastructure conditions [3]. The social impacts are related to traffic accidents [132].

Indicators could measure these impacts to evaluate the improvements in SUFT measures. The continuous evaluation identifies potential problems in SUFT measures for redesigning and correcting these problems. The involvement of stakeholders is crucial in this process.

## 5. Discussion

The proposed conceptual model seeks to holistically represent the relationship between UFT, AS, and TS, including the stakeholders' objectives. Understanding the relationships among stakeholders focuses on one of the main difficulties in public and private

urban planning efforts and in developing public policies due to the conflicting relationships among stakeholders.

A conceptual model may initially appear complex and without functionality. However, conceptual models summarize a system and provide an understanding of that system. In this case, the conceptual model brings knowledge about the UFT phenomenon, promoting a collaborative and participative environment for all stakeholders. Before thinking about technological innovations, it is essential to understand how cargo is generated, moves, and is connected with the city.

The conceptual model does not consider technological innovations since they can change the urban mobility pattern. Thus, providing knowledge about the current status quo of UFT is the primary purpose of this conceptual model. Solutions or measures to reduce environmental, economic, and social impacts need to be measured constantly by indicators, while also evaluating the impacts on stakeholders' objectives. A results-oriented planning process integrating all stakeholders allows assessing each measure's ability to change paradigms, operations, and behaviors.

The limitation to implementing the conceptual model is related to the multidisciplinary team, including all stakeholders. However, as stated by Russo and Comi [133], sustainability depends on monitoring and controlling all steps and costs involved in UFT operation. Thus, it is necessary to develop a governance environment addressing public and private interests [134] in promoting SUFT. In this way, the AS influences (and is influenced by) consumer behavior, which generates freight flows. TS transports the freight flows. As this process occurs many times each day, it is a cyclical and dynamic process that impacts the urban environment for many purposes. Understanding this phenomenon allows us to recognize that the cargo is not isolated in the transport system but is part of the population's wishes and needs. Planning the freight transportation system includes understanding the behavior of stakeholders, their objectives, and thier future perspectives for the development of a sustainable world.

## 6. Practical Implications and Conclusions

The idea that UFT is fundamental to economic development and maintaining the urban lifestyle is widespread and contributes to the increase in congestion, accidents, and the emission of pollutants.

The purpose of this paper was to propose a conceptual model to plan UFT using a holistic view. This holistic approach makes clear the relationship between the AS and TS infrastructure (e.g., proximity to arterial roads, logistical corridors, and ports), land use characteristics (e.g., land costs and land use), sociodemographic characteristics (e.g., population density and number of workers), and protagonism of the stakeholders.

The UFT planning based on this holistic view supports urban planning at an operational, tactical, and strategic level. In addition, the stakeholders' information contributes to collaborative UFT planning, balancing the stakeholders' needs and UFT impacts. The spatial characteristics of the urban area could be a data source for the planning process. In addition to supporting UFT planning, the holistic approach encourages the participation by public managers and inspires a closer relationship with companies to promote SUFT through public policies and, consequently, to promote a sustainable future for cities.

For future works, we suggest proposing indicators to measure UFT impacts for an ongoing planning process. In addition, we suggest the evaluation of the proposed conceptual model through a case study.

**Author Contributions:** Conceptualization, D.R.C. and B.V.B.; methodology, D.R.C. and B.V.B.; validation, B.V.B. and L.K.d.O.; formal analysis, D.R.C.; investigation, D.R.C.; data curation, D.R.C.; writing—Original draft preparation, D.R.C.; writing—Review and editing, D.R.C., B.V.B. and L.K.d.O.; supervision, B.V.B. All authors have read and agreed to the published version of the manuscript.

**Funding:** This research was funded by Coordenação de Aperfeiçoamento de Pessoal de Nível Superior, grant number 306629/2018-6; Conselho Nacional de Desenvolvimento Científico e Tecnológico, grant number 430082/2018-5; Conselho Nacional de Desenvolvimento Científico e Tecnológico, grant number 303171/2020-0, and the APC was funded by Universidade Federal do Ceará and Universidade Federal de Minas Gerais.

**Institutional Review Board Statement:** Not applicable.

**Informed Consent Statement:** Not applicable.

**Data Availability Statement:** Data sharing not applicable.

**Conflicts of Interest:** The authors declare no conflict of interest.

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
