# Peer review of "A Conceptual Model Based on the Activity System and Transportation System for Sustainable Urban Freight Transport"

_sustainability, doi:10.3390/su13105642_

Round 1

Reviewer 1 Report

Referee report

 A Conceptual Model Based on Activity System and Transporta-2 tion System for Sustainable Urban Freight Transport

Introduction

The paper investigates an interesting and noteworthy issue.

Main issues

While potentially interesting, before publication is considered, the Authors should address the following issues:

  • Clarify the research gaps the paper is addressing.
  • Discuss the added contribution to the current state of knowledge.
  • Discuss the policy implications in much more detail w.r.t. the results obtained in the paper.
  • Develop a much more focused literature research on the topic considered since, notwithstanding the bibliographical references used are mostly pertinent, they are not comprehensive nor updated.

Minor issues

  • Deeply revise English
  • Have a native mother tongue speaker help you out in developing a clear and easy-to-read paper

Bibliographical suggestions

Among the missing references to the paper I kindly ask you to consider the following:

Buying online and environmental implications

Bricks or clicks? Consumer channel choice and its transport and environmental implications for the grocery market in Norway, Edoardo Marcucci Valerio Gatta Michela Le Pira Ting Chao Shengnan Li, 2021, cities, Volume 110, Elsevier

Innovation – policy deployment

Planning with stakeholders: Analysing alternative off-hour delivery solutions via an interactive multi-criteria approach, V Gatta, E Marcucci, P Delle Site, M Le Pira, CS Carrocci, Research in Transportation Economics 73, 53-62.

Crowdshipping

1 - Sustainable urban freight transport adopting public transport-based crowdshipping for B2C deliveries, V Gatta, E Marcucci, M Nigro, S Serafini, European Transport Research Review 11 (1), 1-14.

Author Response

Thank you for reviewing our paper. Your comments have enabled us to make an improved version of our work. The amended document accompanies this response; see the red parts in the paper, please.

Reviewer 2 Report

General:

The paper has no scientific outputs. The method of solution is not clearly described, and the benefits of the author (s) are not clear.

A Conceptual Model for Sustainable Urban Freight Transport doesn´t have all elements of Sustainable Urban logistic which are needed for example for the introduction of electric cars and in the future AGV vehicles.  

The Discussion chapter is missing.

The critical assessment of the achieved results is missing in chapter Practical implications and conclusion.

Author Response

(The authors gave the same response as above.)

Reviewer 3 Report

Dear Authors

It was a pleasure to me, reading your manuscript. The issue raised in the manuscript is very interesting and needed. In general, it conforms to the standards of scientific articles.

However, I would like to provide some comments on the content/structure of the manuscript:

  1. I don't see any research methods either in the abstract or in other parts of the article. It should be added. Additionally, some part of the article should describe the research process and/or research methods/methodology.
  2. I do not see any research questions in the manuscript. Because the authors propose „a model addressing sustainable last-mile delivery (...)”, they should create some research questions. It should be added.
  3. I propose a slight modification of the title of the second part of the manuscript: „From city logistics to sustainable urban freight transport -literature review” - this will emphasize the nature of this part.
  4. What research was Figure 1 based on? This is not explicitly mentioned in the article.
  5. Part 3 contains „a conceptual model for sustainable urban freight transport”. There is no detailed description introducing the content of Figure 3. The flows and dependencies presented in Figure 3 require a more detailed explanation.
  6. What are the limitations resulting from the implementation of the proposed model? It should be mentioned in the manuscript.
  7. I do not see discussion of the results.

Author Response

(The authors gave the same response as above.)

Round 2

Reviewer 1 Report

ok

Author Response

We thank you suggestions to improve our manuscript.

Reviewer 2 Report

A Conceptual Model for Sustainable Urban Freight Transport doesn´t have all elements of Sustainable Urban logistic which are needed for example for the introduction of electric cars and in the future AGV vehicles (Figure 2). 

This is not just a technological solution, as the authors stated in the Cover letter, but a change in logistics last mile through the building of urban logistics centers.

Without the urban's logistics centers, we will not be able to take advantage of the current possibilities of electric vehicles due to their reaches, nor will it be able to deploy AGV on a larger scale to deliver shipments.

The paper has the character of a review rather than the article.

Author Response

We thank you suggestions to improve our manuscript. Please see the attachment.
